DATA RELEASE

# First draft genome of the decaploid species, *Ludwigia grandiflora* subsp. *hexapetala*, validated through gene expression

Guillaume Doré[1], Dominique Barloy[1,*,†] and Frédérique Barloy-Hubler[2,*,†]

1 DECOD (Ecosystem Dynamics and Sustainability), Institut Agro Rennes-Angers, INRAE, IFREMER, 35042, Rennes, France
2 UMR 6553 ECOBIO, CNRS, Université de Rennes 1, 35042, Rennes, France

## ABSTRACT

Invasive species are one of the biggest drivers of species extinction. *Ludwigia grandiflora* subsp. *hexapetala* (*Lgh*) is widely invasive in aquatic ecosystems of Europe, North America, and Japan, and also colonizes emergent freshwater soils, but limited genomic data constrain studies of its invasiveness. Here, we report a draft genome assembly of *Lgh*, with a total length of 1.487 Gb, in agreement with the genome size estimated by flow cytometry, despite high fragmentation (111,219 contigs; N50 = 13.5 kb) and low sequencing depth (6.5× Illumina, 1.6× Nanopore). In addition, an analysis combining homology and expression data identified 139,095 protein-coding genes. Moreover, several indicators suggest that the observed fragmentation is largely attributable to unassembled repetitive regions. Thus, despite these limitations, this assembly represents the first genome in the Ludwigioideae subfamily and constitutes a valuable resource for gene discovery, functional genomics, phylogenetic reconstruction, and evolutionary analyses across the Onagraceae family.

**Subjects** Genetics and Genomics, Plants Genetics, Botany

Submitted: 22 July 2025

* Corresponding authors. E-mail: dominique.barloy@agrocampus-ouest.fr, fhubler@univ-rennes1.fr

† Contributed equally.

Preprint submitted at https://doi.org/10.64898/2026.03.09.710498

## INTRODUCTION

Biological invasions are one of the largest threats to biodiversity [1]. Invasive aquatic plants have particularly severe ecological and economic impacts [2]. *Ludwigia grandiflora* subsp. *hexapetala* (*Lgh*), commonly known as the water primrose, is one of the most invasive plants in the world (Figure 1A). Native to South America, *Lgh* was first introduced into France around 1830, likely for ornamental purposes [3]. Since then, *Lgh* has spread rapidly across Europe (Figure 1D) and is now also invading regions in Japan and North America [4–6]. In France, dense mats of *Lgh* (Figure 1C) severely disrupt freshwater ecosystems, causing damage to local biodiversity by obstructing waterways, and interfering with fishing, navigation, and agriculture [7]. Moreover, this amphibious invasive plant has developed the ability to colonize terrestrial environments within its introduction range (Figure 1C) [8]. *Lgh* belongs to the Onagraceae family within the Myrtales order [9]. The Onagraceae family comprises 664 species, and is divided into two subfamilies: Ludwigioideae, represented exclusively by the genus Ludwigia (82 species), and Onagroideae [9]. *Lgh* is decaploid ($2n = 10x = 80$ chromosomes), having arisen through multiple allopolyploidization events (Figure 1B) [10]. Cytogenetic studies revealed that *Lgh*

**Figure 1.** Overview of the amphibious invasive plant, *Ludwigia grandiflora* subsp. *hexapetala* (*Lgh*). (A) Photography of *Lgh* (D. Barloy). (B) Metaphase chromosomes of *Lgh* [10]. (C) Growing ability of *Lgh* in various habitats (D. Barloy). (D) Map of the distribution of *Lgh* in Europe (GBIF), size of points correspond to number recorded.

is an auto-allopolyploid species. Its autotetraploid donor (4*x*) is the diploid species, *Ludwigia peploides* subsp. *montevidensis* (2*n* = 2*x* = 16 chromosomes). *Ludwigia helminthorrhiza* (2*n* = 2*x* = 16 chromosomes) was also identified as a contributor of one genome part (2*x*), while the progenitor(s) corresponding to the last tetraploid (4*x*) genome part remains unknown. The genome size of *Lgh* was estimated to be 1.419 Gb (±0.107 Gb) using flow cytometry [10]. Genomic analyses are essential for deepening our understanding of invasive species' adaptative abilities [11]. However, genomic data for *Ludwigia grandiflora* subsp. *hexapetala* remain scarce, with only the mitochondrial genome and plastome having been recently sequenced and assembled [12, 13]. Furthermore, as of June 2025, only four genomes from the Onagraceae family are available on GenBank, none of which are fully annotated [14]. These include *Chamaenerion angustifolium* and *Epilobium hirsutum*, which are both assembled at the chromosome level, and *Oenothera biennis* and *Clarka xantiana*, which are only available at the scaffold level. Within the Myrtales order, which is composed of about 13,000 species, there are 84 complete or draft genomes, but only four annotated in the RefSeq database.

## CONTEXT

Here, we report the first annotated draft nuclear genome assembly of *Ludwigia grandiflora* subsp. *hexapetala* validated through gene expression, to complement the organelle genomes already available. This genome helps to fill the substantial gap in genomic resources for the Onagraceae family, and more broadly, for the Myrtales order. It also constitutes the first genome reported for the Ludwigioideae sub-family. These genomic resources offer one of the most comprehensive references to date for decoding the acclimation mechanisms of *Ludwigia grandiflora* subsp. *hexapetala* to terrestrial habitats.

**Table 1.** Statistics of sequencing data used for assembling *Ludwigia grandiflora* subsp. *hexapetala* genome.

| Sequencing technology | Sequencing type | Number of reads | Mean size | Total size | Depth |
|---|---|---|---|---|---|
| Illumina MiSeq | Short reads | 2 × 23,067,490 | 207.95 | 9,606,411,104 | 6.5× |
| Oxford Nanopore | Long reads | 482,619 | 4944.6 | 2,386,362,939 | 1.6× |

Given the scarcity of genomic resources for the Myrtales order, especially within the Onagraceae, this new genome paves the way for comprehensive studies on genetic diversity, phylogenetics, and evolutionary dynamics within this complex family.

## MATERIAL AND METHODS

### Plant material

For DNA analysis, *Ludwigia grandiflora* subsp. *hexapetala* (*Lgh*) plants were collected from the Mazerolles swamps (N47 23.260, W1 28.206) near Nantes (France). Several plants were grown in a growth chamber from a single 10 cm stem, using vegetative propagation in order to avoid potential genetic variation, as described in Barloy-Hubler *et al.* [13]. For RNA analysis, *Lgh* plants were collected from multiple points within the Mazerolles swamps (N47 23.260, W1 28.206) near Nantes, France. These plants were kept either in aquatic (submerged plants) or terrestrial (emerged plants) conditions for six months in a greenhouse. After 6 months, 10 cm from the apex of *Lgh* plants were cut. These fragments were put in terrestrial conditions for 13 days in growth chamber.

### DNA and RNA extractions and sequencing

*Lgh* is a plant that contains high levels of polyphenols and tannins in its tissues. As all recalcitrant plants, DNA and RNA extractions were challenged and required to adapt DNA and RNA usual protocols, especially to performed high molecular weight (HMW) DNA extraction required for long reads sequencing [15]. High-molecular-weight (HMW) DNA was extracted from new buds following the protocol described in Barloy-Hubler *et al.* [13]. DNA libraries were then sequenced using Illumina MiSeq (RRID:SCR_016379) and GridION Oxford Nanopore technologies (RRID:SCR_016379) to obtain short (SR) and long (LR) reads, respectively (Table 1). The sequencing was carried out at the Genome Transcriptome Facility of Bordeaux (INRAE-UMR 1202 BIOGECO, Bordeaux, France) in 2020.

RNA from roots and aerial parts (stems and leaves) were extracted separately from a pool of six plants with three biological replicates. We used a protocol especially adapted for recalcitrant species [16] following by a cleaning step using Nucleospin RNA kit (Macherey Nagel). These RNA were then sequenced at the Montpellier GenomiX (MGX) platform in 2020. Two sequencing runs of Illumina NovaSeq (2 × 150 bp) and one run of Illumina MiSeq (2 × 300 bp) were done.

#### *De novo genome assembly*

The pipeline used is described in Figure 2. Sequence quality controls have already been described in Barloy-Hubler *et al.* [13]. In short, LR and SR were filtered with Guppy v4.0.14 (RRID:SCR_023196) and fastp v0.20.0 (RRID:SCR_016962), respectively [17]. We performed a hybrid *de novo* genome assembly by integrating both long-read (Nanopore) and short-read (Illumina) sequencing data. Long reads were first error-corrected using Ratatosk v0.9.0 (RRID:SCR_021824), which leverages short-read Illumina data to reduce long-read error rates sixfold compared to raw reads [18]. Reads corresponding to chloroplast and

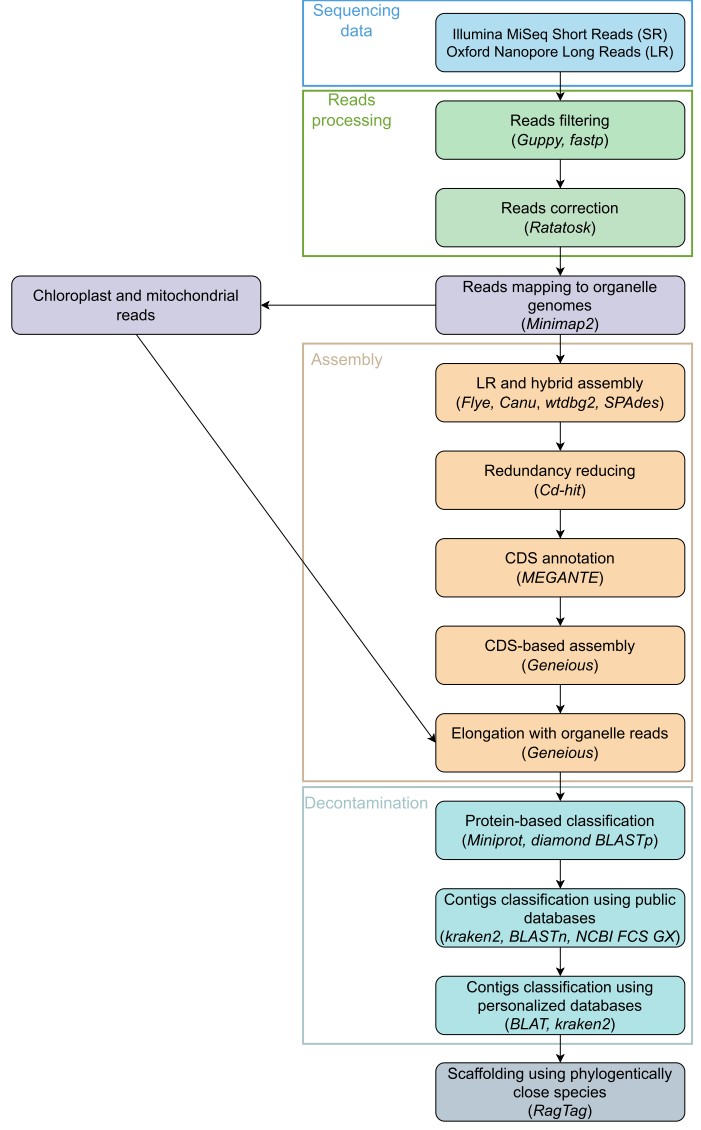

**Figure 2.** Pipeline used for *Ludwigia grandiflora* subsp. *hexapetala* genome assembly.

mitochondrial genomes were initially depleted by mapping against reference sequences using minimap2 v2.15 (RRID:SCR_018550) [19]. Next, three long-reads assemblers were employed in parallel to maximize contiguity and assembly accuracy: Flye v2.9.1 (RRID:SCR_017016), Canu v2.2 (RRID:SCR_015880) and wtdbg2 v2.5 (RRID:SCR_017225) [20–22]. A complementary hybrid assembly was realized using SPAdes v3.15.5 (RRID:SCR_000131) [23]. Subsequently, we scaffolded and merged individual assemblies in order to resolve conflicts and select the highest-quality contigs and scaffolds from each tool. This multi-assembler strategy allowed us to exploit the strengths of each algorithm, resulting in a more complete and accurate nuclear genome assembly. Finally, Cd-hit v4.8.1 (RRID:SCR_007105) (parameters: -c 0.90) was used to reduce redundancy among contigs [24].

## CDS annotation

To aid assembly of the remaining contigs, CDS were annotated by MEGANTE (release 2018-02), using *Arabidopsis thaliana* as the gene prediction reference [25]. Genes with the exact same sequence were considered as duplicated. Amino acid sequences predicted by MEGANTE, but lacking functional annotation were subsequently aligned against the NCBI non-redundant (NR) database for functional inference (28-07-2023) using diamond BLASTp v2.1.11 (RRID:SCR_016071) (parameters: –max-targets-seqs 1) on the Europe Galaxy server (RRID:SCR_006281) [26, 27]. All ORF were also annotated by InterProScan v5.59-91.0 (RRID:SCR_005829) [28]. All matching GO terms were submitted to ReViGo v1.8.2 (RRID:SCR_005825) with their occurrence frequency in *Lgh* genes (Parameters: Resulting list size = "Medium (0.7)", "values associated with GO terms represent" = "Higher value is better", Database = "*Arabidopsis thaliana*") [29]. ReViGo results were visualized as representative clusters via the TreeMap method, where the size of each rectangle is based on the occurrence frequency of each term in the *Lgh* genome. Proteins were classified as having uncharacterized proteins if all associated annotations contained at least one of the terms: "hypothetical", "uncharacterized", "detected", "unknown", "predicted protein" or "unnamed". WoLF PSORT (RRID:SCR_002472) was used to predict subcellular protein localization, keeping the best match for each protein [30]. LOCALIZER v1.04 was used to verify subcellular localization predictions for mitochondrial, chloroplast and nuclear proteins made by WoLF PSORT [31]. Original predictions were kept if the two software agreed or if only WoLF PSORT had a prediction. GhostKOALA v3.1 assigned KEGG annotations to each ORF [32]. KEGG Mapper v5.2 was used to represent the metabolic pathway maps, and chloroplast and mitochondria coded proteins were added [33].

## Biocuration

All CDS were extracted and *de novo* assembled using Geneious Prime "*de novo* assembly" option (version 2025.0.3, RRID:SCR_010519) (parameters: Minimum Overlap: 100, Minimum overlap identity: 95%, other parameters were the same as the "Low Sensitivity/Fastest"). These assemblies were used to align all genome contigs to identify potential overlaps and enable their further scaffolding using the "Map to Reference(s)" function (parameters: Minimum Overlap: 100, other parameters were the same as the "Low Sensitivity/Fastest") from Geneious. Contigs mapped to the same CDS were then assembled using the "*de novo* assembly" function (same parameters as before) from Geneious. As plant nuclear genomes contain mitochondrial and chloroplast sequences [34], long reads and assemblies belonging to the mitogenome and plastome were subsequently reintegrated into the assembly process to evaluate whether they could aid in scaffolding, and improving contig connectivity. Contigs that were elongated by the first step were *de novo* assembled with the reads mapped to them (same parameters as before). Elongated contigs after this second step were all *de novo* assembled (same parameters as before). Finally, all contigs were mapped to the new elongated contigs with "Map to Reference" function (same parameters as before).

## Decontamination

Decontaminating the genome assembly is essential for removing foreign DNA sequences, such as those from bacteria, fungi, or environmental contaminants, which can lead to misannotations and erroneous conclusions. To remove contaminant contigs, all amino acid sequences predicted by MEGANTE were mapped back to the final contigs using Miniprot

v0.13 (parameters: –outc 1 –outs 1) [35]. Each mapped predicted protein was assigned to a phylum via diamond BLASTp. Those classified within Viridiplantae (Streptophyta and Chlorophyta Phyla) were retained as non-contaminants, whereas proteins matching other phyla were flagged as contaminants. Accordingly, any contig harboring at least one contaminant predicted protein, or predominately comprising unannotated proteins (at least 17 out of 20) were then classified by kraken2 v2.1.3 (RRID:SCR_005484), as well as contigs that didn't contain any protein [36]. Contigs classified as non-viridiplantae were compared to the nt database, and BLASTn v2.16.0 (RRID:SCR_004870) [37]. NCBI FCS GX v0.5.5 (RRID:SCR_026367) (parameters: –tax-id 33090) was used on all contigs to check for contamination [38]. Contigs that were still not classified were then compared to other Myrtales species genomes from EnsemblPlants with BLAT v.39x1 (RRID:SCR_011919) [39, 40]. Those matching with at least two species were conserved, while the others were sent to Kraken2 classification using the UniVec_Core and four Onagraceae homemade database (*Chamaenerion angustifolium* (Accession: GCA_946814005.1), *Oenothera biennis* (Accession: GCA_025802395.1), *Epilobium hirsutum* (Accession: GCA_965153245.1), and *Clarkia xantiana* (Accession: GCA_036873135.1)). Contigs classified as Onagraceae were kept.

## Using RNA-Seq data to correct CDS annotations

Raw RNA reads were trimmed by Cutadapt v1.15 (RRID:SCR_011841) (parameter: -m 2) (Martin 2011). Fastp v0.23.4 was used to correct and merge reads (parameters: –disable_adapter_trimming –dedup -l 6 –trim_poly_g –trim_poly_x –average_qual 20 –cut_right –cut_right_mean_quality 20 -c –overlap_len_require 10 (for NovaSeq reads) or 20 (for MiSeq reads) -m) (Chen 2023). Merged and unmerged read were then mapped to the concatenated sequence (gap of 100 Ns between sequences) of *Lgh* genome (including plastome and mitogenome) using Geneious Prime "Map to Reference(s)" function (parameters: Mapper: "Geneious RNA"; Sensitivity: "Low Sensitivity/Fastest"; Map multiple best matches: "To all"). RPKM was calculated for each protein-coding gene. Gene annotations with a RPKM < 0.1 (value recommended to filter noise from real expression [41]) and marked as "hypothetical protein" (with no homology in databases) were deleted.

## Non-coding genomic features annotation

Ribosomal and transfer RNA were respectively annotated by pybarrnap v0.5.1 (Parameters: –kingdom euk –reject 0.8) and tRNAscan-SE v2.0.12 (RRID:SCR_008637) [42, 43]. DANTE v0.1.9 predicted transposable elements in the *Lgh* genome using the database "Viridiplantae_v3.0" [44]. Other repeats were annotated by RepeatMasker v4.1.8 (RRID:SCR_012954) [45] and Tandem Repeats Finder v4.09.1 (RRID:SCR_022065) [46].

## Scaffolding

RagTag v2.1.0 was used to scaffold the *Lgh* genome (Parameters: –mm2-params "-x asm20") [47]. Three nuclear reference genomes were used iteratively: *Chamaenerion angustifolium*, *Epilobium hirsutum* and Punica granatum (see Supplementary Table 1 for accessions and details). We initially scaffolded the genome using *Chamaenerion angustifolium*, as it yielded the highest number of mapped *Lgh* contigs. We used *Epilobium hirsutum* as a second scaffold, due to it being phylogenetically closer to *Lgh*. This improved assembly coherence compared to *Punica granatum* [9, 48]. We applied this multi-reference

scaffolding strategy because no fully assembled genome existed for any species that was sufficiently close to *Lgh* phylogenetically [49].

### Estimation of chromosome size

A study compiling 419 angiosperm species showed that the average length of a chromosome (*L*, in μm) follows a power law with respect to the amount of chromosomal DNA (*C*, in picograms): $L = a \cdot C^a$ with $a \approx 4.64$ and $a \approx 0.506$ [50]. This relationship can be inverted to estimate *C* from *L*: $C = (L/4.64)^{1/0.506}$ then converted to bases with the following reference conversion: $1 \text{ pg} \approx 0.978 \times 10^9$ bp [51]. Using these formulae, *Lgh*'s chromosome size range was estimated to be approximately 30 to 134 Mb per chromosome, based on chromosome length measured by cytological observations between 0.8 and 1.7 μm [3]. This is greater than the estimated average chromosome size of 17.7 Mb, corresponding to the size of the decaploid genome ($\approx$1.42 Gb) divided by the number of chromosomes ($2n = 10x = 80$). Estimates of genome size by flow cytometry are approximate and subject to variations estimated at ±29% or more, depending on the internal standard and specific experimental conditions (extraction buffer, fluorochrome, plant tissue/metabolites or tissue type) [52]. This correction would take us from a minimum size of 21.3 Mb to a maximum size of 172.9 Mb.

### Qualitative and quantitative genome evaluation

BUSCO v5.8.0 (RRID:SCR_015008) was used to evaluate the quality of the assembly [53]. Long reads and short reads were mapped to the genome using Minimap2 v2.15 (RRID:SCR_018550) (Parameters: -x map-ont for LR and -x sr for SR) [19]. The statswrapper script from BBMap (RRID:SCR_016965) calculated genome statistics (N50, L50, N90 and L90) [54]. Meryl v1.4.1 (RRID:SCR_026366) (Parameters: Estimate the best k-mer size; Genome size: 1419000000) [55] was used to count k-mer in short reads. Then, Merqury v1.3 (RRID:SCR_022964) [55] compared k-mers present in *Lgh* genome assembly and in unassembled reads. LTR_retriever v3.04 (RRID:SCR_017623) was used to calculate the LAI (LTR Assembly Index)  [56]. LTR_finder v1.07 (RRID:SCR_015247) predicted LTR-RTs used in LTR_retriever [57].

### Orthologous genes analysis

Proteomes from 12 species, corresponding to all nine Myrtales order species with annotated genomes, and three Malvid clade species used as outgroups, were downloaded from NCBI Genbank (see Supplementary Table 1 for accessions and details). *Chamaenerion angustifolium* and *Epilobium hirsutum* genomes were annotated by the Helixer Web Service v0.3.4 using only the chromosomes [58]. To find common orthogroups and orthologous genes between *Lgh* and other species, these 15 proteomes (the 12 downloaded from NCBI Genbank + *C. angustifolium*, *E. hirsutum* and *Lgh*) were analyzed by Orthofinder v3.0.1 (RRID:SCR_017118) [59]. The R package UpSetR (RRID:SCR_026112) was used for the upset graph [60].

### Genome synteny

LASTZ v1.04.15 (RRID:SCR_018556) in Geneious was used to compare synteny between *Lgh* and *Epilobium ciliatum* (GenBank accession: GCA_965637465.1; published in July 2025) genomes [61]. *Lgh* scaffolds bigger than 1 Mb, and *E. ciliatum* chromosomes were used. Sequence alignments of at least 1 Mb in total were conserved.

**Table 2.** *Ludwigia grandiflora* subsp. *hexapetala* genome statistics.

| Statistics | Data |
|---|---|
| Number of contigs and scaffolds | 111,219 |
| Number of scaffolds | 50 |
| Number of contigs | 111,169 |
| Total size (bp) | 1,486,583,937 |
| Maximum scaffold length (bp) | 53,046,275 |
| Minimum contig length (bp) | 4,970 |
| N50 (bp) | 13,498 |
| L50 | 26,764 |
| N90 (bp) | 6,387 |
| L90 | 91,755 |
| GC content (%) | 39.7 |
| Long reads mapped (%) | 99.65 |
| Short reads mapped (%) | 98.62 |
| BUSCO score (%) | C:96.3 [S:15.5, D:80.8], F:1.6, M:2.1 |
| K-mers completeness (%) | 99.5 |

## RESULTS AND DISCUSSION

### Sequencing, assembling and scaffolding

Following *de novo* assembly, we obtained 175,249 initial contigs. To reduce redundancy, we applied CD-HIT-est at a 90% identity threshold, yielding 158,330 non-redundant contigs. Next, contigs sharing overlapping coding sequence (CDS) annotations were merged, resulting in a refined set of 149,835 contigs. To improve scaffold continuity, we incorporated long reads along with plastome and mitogenome assemblies, which slightly increased contig contiguity, producing 149,534 contigs. Finally, after comprehensive decontamination removing bacterial, fungal, and other non-target sequences, we retained 117,972 high-confidence nuclear contigs for downstream analysis. These contigs were iteratively scaffolded against *C. angustifolium*, *E. hirsutum* and *P. granatum*. Initially, 3088 contigs aligned to the *C. angustifolium* genome (Onagraceae), producing 49 scaffolds. These scaffolds, along with any remaining contigs, were then mapped to the *E. hirsutum* genome, resulting in 2,260 alignments and the formation of 32 additional scaffolds. In the final step, 1,550 scaffolds and unscaffolded contigs were aligned to the *P. granatum* genome (Punicaceae), generating 50 new scaffolds. Overall, this multi-tiered scaffolding strategy reduced the total number of nuclear contigs from 117,972 to 111,219 (Table 2), creating a final assembly of 111,219 scaffolds. This elevated number of scaffolds likely results from the limited lengths of ONT long reads, which were insufficient to span repetitive regions. As a consequence, many repeats remain unresolved during assembly, leading to a highly fragmented genome that could not be assembled into chromosome-scale scaffolds (Figure 3).

Indeed, obtaining high-quality, high molecular weight (HMW) DNA in plants is critical for successful long-read sequencing. However, plant genomes such as *Lgh* often pose significant challenges due to the presence of secondary metabolites such as polysaccharides and polyphenols, which can co-purify with DNA, and cause fragmentation, or inhibit downstream enzymatic reactions [62]. Given that our efforts to extract high molecular weight (HMW) DNA from *Lgh* have been unsuccessful, and the DNA quality remains suboptimal, alternative approaches such as stLFR or TELL-Seq could represent robust solutions for obtaining chromosome-scale genome assemblies. Rather than depending on ultra-long intact DNA fragments, these synthetic long-read technologies barcode

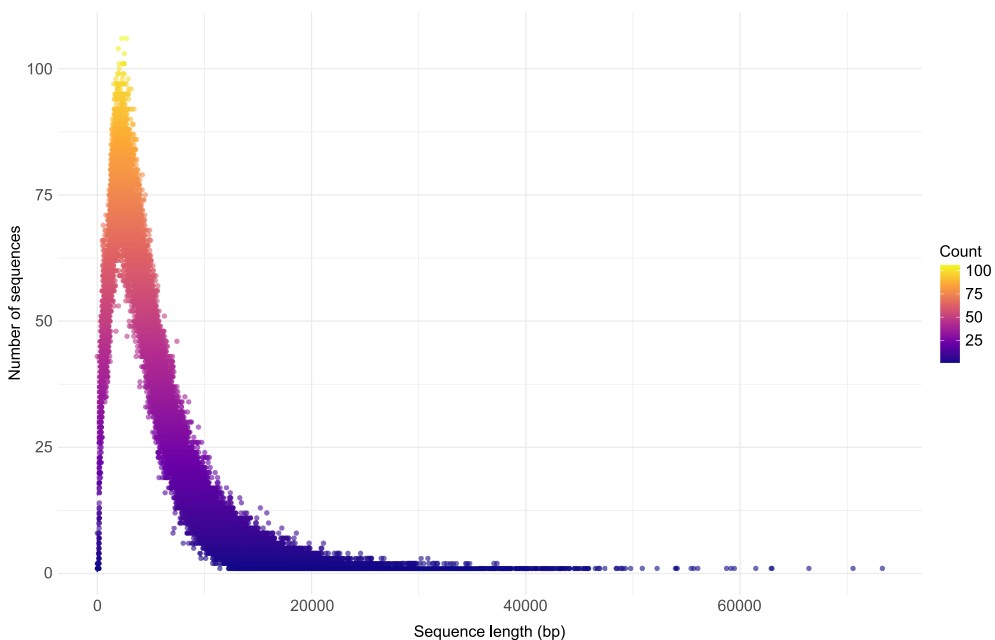

**Figure 3.** Distribution of Oxford nanopore long read lengths in *Ludwigia grandiflora* subsp. *hexapetala* sequencing.

fragmented DNA molecules within a single tube, enabling the in silico reconstruction of long-range genomic information (≃10–100 kb) from standard short-read sequencing data. These methods have been successfully applied in plants; for example, TELL-Seq has demonstrated effective de novo genome assembly, and accurate structural variant detection in *Arabidopsis thaliana* [63]. Moreover, given that *Lgh* exhibits decaploidy (10×), employing capture sequencing (Hi-C, Pore-C, Omni-C) can be considered, as this technique has been successfully applied to complex polyploid species such as *Brassica napus* and *Eutrema japonicum* [64, 65].

The presence of cell metabolites such as lignin, phenolics, alkaloids, terpenes, and flavonoids in *Lgh* tissues represent significant challenges to generate DNA suitable for production of long reads [15]. The difficulty encountered during Lgh DNA extraction also resulted in poor sequencing depth especially for Oxford Nanopore reads (1.6×). It can explain the high fragmentation observed in *Lgh* genome assembly and the absence of chromosome-level scaffold.

## Genome qualitative and quantitative assessment

Different statistics were used to assess *Lgh* genome assembly (Table 2). The assembled genome has a total length of 1.487 Gb, which falls within the range of the genome size estimated by flow cytometry (1.419 ± 0.107 Gb) [10]. Scaffold lengths range from 5 kb to 53 Mb. Based on calculated chromosome size estimates, ranging from 21.3 Mb to 172.9 Mb, four scaffolds (from 35.2 to 53 Mb) fall within the expected range for complete chromosomes (out of the 40 expected). The biggest scaffold size is bigger (53 Mb) than *C. angustifolium* and *E. hirsutum*'s biggest chromosomes (46 Mb and 22 Mb, respectively). Furthermore, five other scaffold sizes are between 15 and 46 Mb long, and thereby within

the range of those two species chromosome lengths, suggesting these scaffolds might be complete or almost complete (Supplementary Figure 1). The N50 value is 13,498 bp which shows that there are still a lot of very small contigs in the assembly. Almost all LR and SR (99.65% and 98.62%, respectively) are mapped on *Lgh* genome. The BUSCO score is 96.3% showing a good completeness of the genome. The presence of 80.8% of duplicated BUSCOs in the *Lgh* genome is not surprising for this decaploid species, as the octoploid species *Cardamine chenopodiifolia* presented 80.9% duplicated BUSCOs [66]. BUSCO score is a metric based on gene space which is often insufficient to evaluate the completeness of genome, especially in repeat rich plant genomes [56, 67]. Merqury showed that 99.5% of k-mers present in reads were also present in *Lgh* genome. This also demonstrate a good completeness of the genome. The LAI calculation by LTR_retriever couldn't be performed as the intact LTR-RT content was too low (0.06%), the minimum required being 0,1%. This result may show that LTR-RT were not completely assembled in *Lgh* genome. Thus, this may participate in the assembly fragmentation.

As a conclusion, the final *Lgh* assembly comprises 111,219 contigs with an N50 of 13.5 kb, reflecting a low sequencing depth (6.5× Illumina, 1.6× Nanopore) and the inherent complexity of this polyploid plant genome. However, multiple lines of evidence, including BUSCO and k-mers analysis, indicate that this fragmentation does not compromise the completeness of non-repetitive genomic regions.

## Protein-coding genes annotations

MEGANTE found 246,242 protein-coding genes (PCG) in the *Lgh* genome. More than half of them (126,744) were annotated as "hypothetical protein" with no homology in databases. To filter these genes, RNA-Seq data were used. A total of 19,597 (15.5%) of these genes showed evidence of expression (RPKM > 0.1) and were retained, together with genes displaying homology to entries in public databases, resulting in 139,095 final gene annotations in the *Lgh* genome (Supplementary Table 2). This total is much bigger than other related species such as *Arabidopsis thaliana*, which had 48,265 PCG and *Punica granatum* which had 36,608 PCG (See Supplementary Table 1 for accession). However, *Lgh* is a decaploid species and these two species are diploid, which might explain this difference, as polyploid species tend to have more PCG [68]. The decaploid genome of *Houttuynia cordata* ($2n = 10x = 90$), with a genome size of 2.63 Gb, contains only 139,087 PCG, which is almost equal to *Lgh* PCG number [69]. *H. cordata*, like *Lgh*, is reported to be an autoallopolyploid species. However, *H. cordata* only has two ancestral parents, with one parent contributing 9× parts of the 10x genome, whereas the two *Lgh* donors identified contribute to 4x and 2x of 10x genome [10]. Another example is the *Prunus domestica* genome, which is a hexaploid species, possessing 138,361 PCG with a genome size of 1.3 Gb [70]. All these data suggest that although the number of genes can be linked to genome size and/or polyploidy, it is not really consistent between species and can vary a lot.

Among the 139,095 PCG annotated in the *Lgh* genome, 91.4% (111,530) of them are in single copy while 8.6% (10,529) are duplicated (Supplementary Table 2). This gave a total of 122,059 unique genes in the *Lgh* genome. Among them, 87% (106,292) had at least one of diamond BLASTp, GO, Interpro, or KEGG annotation (Figure 4). The other 15,767 proteins are considered as hypothetical proteins since no homology was found with any sequences in the databases. Among the annotated proteins, almost 50.7% (53,883) were annotated as unknown function proteins. The observation that fewer than 9% of genes are identified as



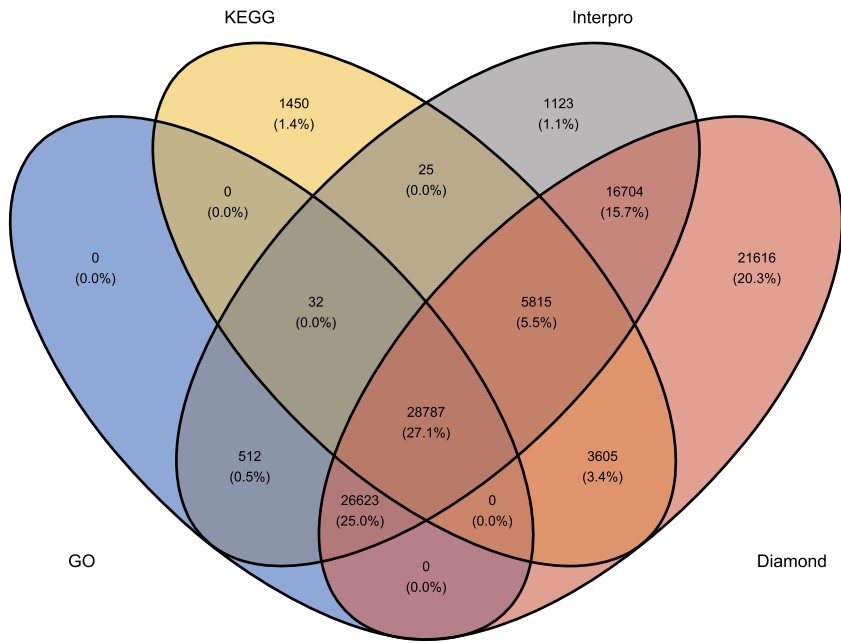

**Figure 4.** Venn diagram of *Ludwigia grandiflora* subsp. *hexapetala* genome annotations : KEGG (yellow), Interpro (grey), GO (blue) and diamond (red).

duplicated in *Lgh* decaploid genomes may be attributed to several factors. Firstly, many duplicated genes undergo rapid silencing through mechanisms such as reciprocal silencing, where redundant gene copies are epigenetically silenced without affecting the organism's phenotype. This silencing can occur shortly after polyploidization, leading to the apparent absence of duplicated genes in functional analyses [71]. Secondly, duplicated genes often experience functional divergence through processes like subfunctionalization and neofunctionalization. In subfunctionalization, each gene copy retains a subset of the original gene's functions, while in neofunctionalization, one copy acquires a new function. These processes can make it challenging to identify duplicated genes using standard annotation methods, as the functional roles of the duplicates may differ from the original gene [72].

## Protein-coding genes functional characterization

GO terms were assigned to 55,954 proteins. Analysis of these GO terms according to the number of occurrences show that key molecular functions such as oxidoreductase activity, nucleic acid binding or protein binding were overrepresented (Supplementary Figure 2). Biological process GO annotation such as protein phosphorylation, transmembrane transport or proteolysis were very present in *Lgh* proteins (Supplementary Figure 3). GO terms for cellular components such as the membrane, nucleus and ribosome particularly stood out (Supplementary Figure 4). Some of the most frequent keywords in all GO terms are process, exact or activity (Figure 5). Protein subcellular localizations were predicted by WoLF PSORT and LOCALIZER (Figure 6, Supplementary Table 3). Nucleus is the most common localization with 42,009 proteins (30.2%), followed by the chloroplast with 32,976 (23.71%) proteins, then the cytosol with 30,022 proteins. As chloroplasts host various

cells h2o result molecules response movement rate hydrolysis extent complex formation amino metabolic group state cellular frequency expression chemical results specific activity membrane diphosphate process via catalysis pathways compound exact directed stimulus structure acceptor cell transfer biosynthetic organism acid form catabolic modulates side dna breakdown acids phosphate linked atp binding groups reactions regulation contains enzyme narrow molecule change transporter synthesis related transmembrane rna between assembly proteins reduces broad production transport reaction

**Figure 5.** REVIGO word cloud of frequent GO term keywords in *Ludwigia grandiflora* subsp. *hexapetala* genome, based on occurrence frequency.

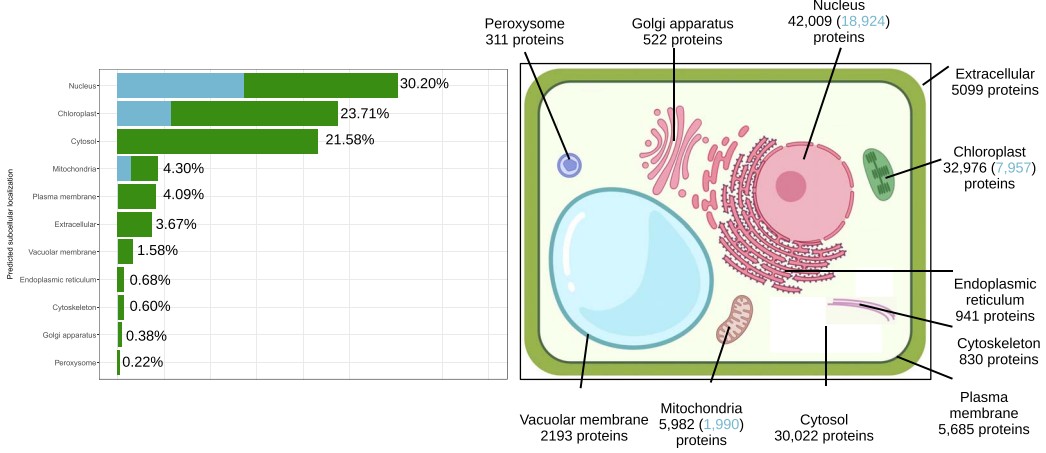

**Figure 6.** *Ludwigia grandiflora* subsp. *hexapetala* nuclear-encoded proteins subcellular localization prediction by WoLF PSORT and LOCALIZER. Blue color in both the graph and the schema refers to localizations supported by concordant predictions from the two algorithms, whereas green color represents localizations predicted solely by WoLF PSORT.

important processes such as photosynthesis, it is not surprising to find a high proportion of proteins here [73]. Plastids are morphologically and functionally diverse organelles that are dependent on nuclear-encoded, plastid-targeted proteins for all biochemical and regulatory functions. In plant genomes, the proportion of proteins predicted to be localized in the plastids typically ranges from 15–25%, with notable variation among species: in tomatoes (*Solanum lycopersicum*), a comprehensive plastid protein atlas predicted approximately 7,473 nuclear-encoded plastid proteins, which represents about 22% of the proteome [74]. In *Arabidopsis thaliana* and other angiosperms, genome-wide analyses predict 3,000–5,000 plastid-targeted proteins, accounting for roughly 15–20% of total proteins [75]. A 23.7% prediction for *Lgh* is consistent with known values for plant genomes. The use of two prediction tools permitted to correct first prediction by WoLF PSORT which attributed 31.6% of proteins to chloroplast. LOCALIZER can only predict localization in 3 subcellular compartments (nucleus, chloroplast and mitochondria). It is recommended to use multiple tools for a better robustness of predictions [76]. However, very few tools can handle prediction for more than 100,000 proteins at once. Subcellular localization predictions in this study should be taken with caution especially for those made only by WoLF PSORT.

What is more surprising is the difference between Cellular Component GO terms and the predicted localization. The GO term "membrane" (GO:0016020) is a broad parent category that encompasses various membrane types like thylakoid, mitochondrial, nuclear, endoplasmic reticulum membranes. Thus, a protein localized in the chloroplast,

mitochondrion, or nucleus membrane can still carry the general annotation "membrane" in its GO terms. KEGG annotations were assigned to 39,714 genes and important metabolic pathways such as the glycolysis and TCA cycle appear to be complete (Supplementary Figure 5). Annotating ≃33% of proteins in the *Lgh* genome with KEGG is relatively high, as using KEGG-based tools, the prediction reaches 26% for *Brassica napus* [77]. This result can demonstrate the good quality of annotations in *Lgh* genome.

### Non-coding RNA genes annotation

The *Lgh* genome comprises a total of 15,834 rRNA genes, including 2,018 5S, 3,939 5.8S, 6,237 18S and 3,640 28S rRNA genes (Supplementary Table 2). The genome of *Ludwigia grandiflora* subsp. *hexapetala* harbors a number of rRNA genes that is notably high compared to other plant species, even though this number can be very variable in plants (generally from 500 to 40,000) [78], and for the 5S rRNA gene, it can even reach 75,000 copies [79]. This abundance may be attributed to the species' polyploid nature and the presence of multiple ribosomal DNA (rDNA) loci. Usually, rRNA genes are organized in two repeated units: the 35S rDNA, which contains the 18S, 5.8S and 28S rRNA genes, and is 8 to 12 kb long, and the 5S rDNA, which contains only the 5S rRNA gene, and is about 800–900 pb long [80]. Both rDNA units are often grouped in clusters. The 35S rDNA clusters are long tandem repeats localized in nucleolar organizer regions (NORs) [79, 81]. In the *Lgh* genome, 17 clusters of 35S rDNA of more than 50 kb were found. However, NORs can span millions of base pairs, suggesting that these high repetitive regions might be incomplete in the *Lgh* genome [82]. Even though they are incomplete, we can expect the presence of multiple NORs, which are often associated with polyploidy and genomic redundancy [83]. Such a high copy number of rRNA could have implications on ribosome biogenesis and cellular metabolism, potentially enhancing the plant's adaptability to various environmental conditions. A total of 8,417 tRNA genes were annotated in the *Lgh* genome, surpassing the 6,475 tRNA genes identified in *Ipomea nil*, the plant species with the highest number of tRNA genes out of 128 others in a study [84]. In plant genomes, the most abundant tRNA isotypes are tRNA-Ala, tRNA-Pro, tRNA-Ser, tRNA-Arg and tRNA-Leu [85]. Three of them were the most abundant tRNA isotypes in the *Lgh* genome (tRNA-Arg, tRNA-Ser and tRNA-Leu), with frequency comprised between 8.5 to 9.2%, representing 26.6% of total tRNA number (Supplementary Figure 6). All anticodons were present in *Lgh* tRNA genes, although some of them very underrepresented, such as Pro-GGG (0.07%), Gly-ACC (0.07%), Leu-GAG (0.04%) and Tyr-AUA (0.04%). This was unsurprising, as these anticodons are among the least abundant in the Plant Kingdom (from 0.009 to 0.038%) [84]. On the other hand, Met-CAU (7.6%), Asn-GUU (7.5%), Arg-ACG (4.8%) and Asp-GUC (4.3%) were overrepresented in *Lgh* tRNA genes. The three anticodons Met-CAU, Asn-GUU and Asp-GUC are the three most abundant in plants (from 4.02 to 5.04%) [84].

### Repeated elements

The *Lgh* genome is made up of 8.5% (126 Mb) of repeated elements (Figure 7, Supplementary Table 4). A total of 625,820 simple sequence repeats, 388,071 tandem repeats and 140,494 transposable elements were identified. Among the transposable elements, almost 70% (98,220) were LTR Ty3-gypsy. The proportion of repeated elements in the *Lgh* genome was very small compared to other polyploid plant genomes, such as the decaploid species *H. cordata* (55.52%), or the tetraploid species *Camelia granthamiana* (68.48%) genomes [69, 86].

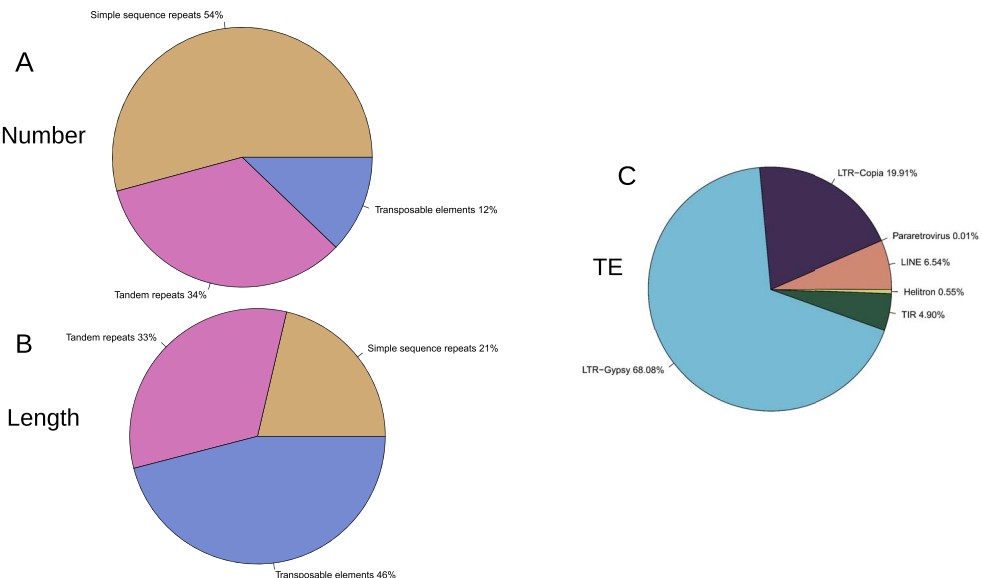

**Figure 7.** Repeated elements in *Ludwigia grandiflora* subsp. *hexapetala* genome. (A) Distribution of number of repeated element types. (B) Distribution of sizes of repeated element types. (C) Distribution of number of transposable elements major families. Detailed number can be found in Supplementary Table 4.

This low rate of *Lgh* repeated elements (8.5%) could simply be due to the fact that our analysis did not assemble them and that they are therefore missing. LTR elements, frequently span 4–31 kb and are often nested within each other, as observed in maize or wheat, with clusters of up to 155 kb [87, 88]. This produces contig breaks where reads cannot span unique flanking regions. This phenomenon results in collapsed or missing copies, and as repetitive regions are not properly assembled, they are underrepresented or completely missing in the final genome. Another hypothesis for the repeat content in the *Lgh* genome is the low percentage of repeat content in some polyploid species compared to diploid species [89]. This observation can be explained by a purging mechanism of repeated elements, and in particular LTRs, following whole genome duplication [90].

## Orthologs comparison in the Malvids clade

The *Lgh* proteome was compared with those of 14 Malvid species using OrthoFinder to identify orthologous gene sets. This analysis revealed that 9,663 orthogroups are shared across all species, illustrating a substantial core proteome conserved within the clade (Figure 8A). However, 7,090 orthogroups were unique to *Lgh*, which is a lot more than in any other analyzed species. Almost 23% (28,746/125,181) of genes assigned to orthogroups in *Lgh* were species-specific, which is a bigger proportion than in any other studied species (from 0.5% in *R. argentea* to 12.5% in *A. thaliana*) (Figure 8B, Supplementary Table 5). Orphan or lineage-specific genes, defined as genes lacking detectable similarity to genes in other taxa, constitute a variable but meaningful subset of plant genomes. Previous reviews report that orphan genes typically represent a minority of gene content, up to ≈30% when using robust annotation strategies [91, 92]. This result is also in agreement with the specificity of the *Ludwigia* genus which is the only genus available for the Ludwigioideae subfamily, and is phylogenetically distinct from all other species of Onagraceae family [48].

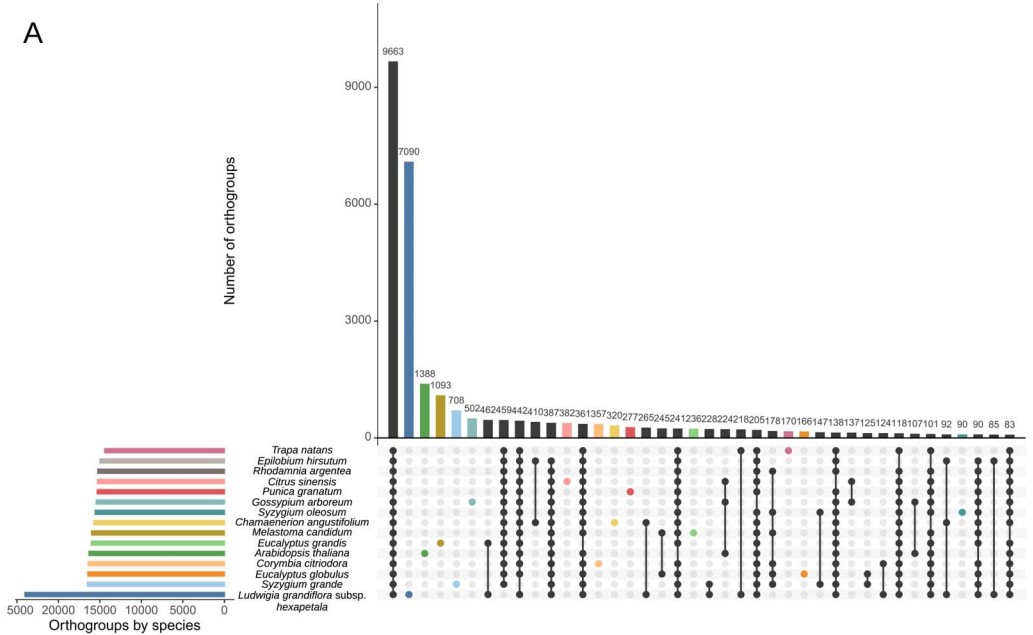

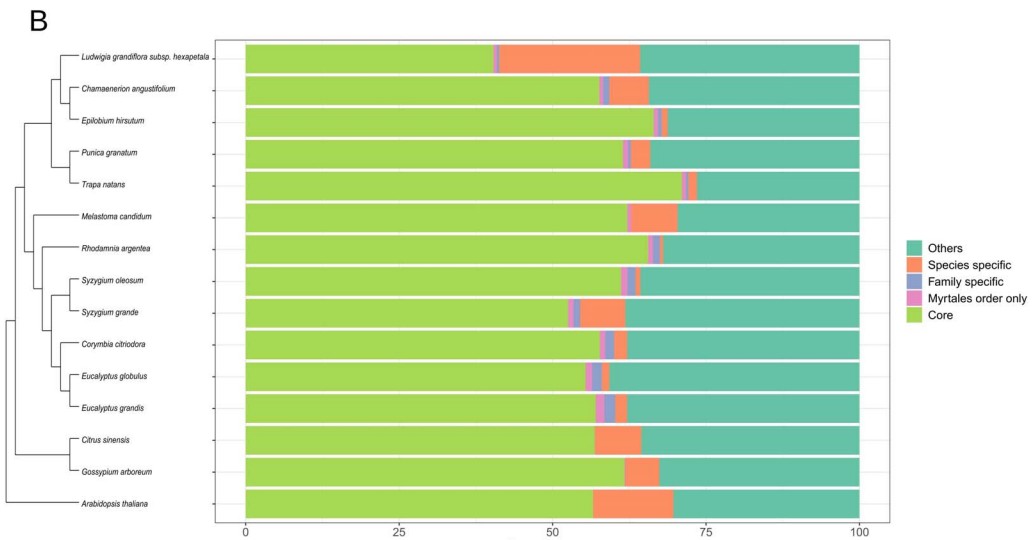

**Figure 8.** Orthologs comparison of Malvid species proteomes. (A) UpSet plot of orthogroups shared among different species, sorted by frequency. (B) Phylogenetic tree based on all genes and percent of genes classified in orthogroups shared or unique among species. Core refers to genes in orthogroups present in all species. References for genomes used in this analysis can be found in Supplementary Table 1. Detailed number are in Supplementary Table 5.

This prevalence of orphan genes in the *Lgh* polyploid invasive plant may contribute to its adaptability and invasive success. In polyploid plants, whole-genome duplications (WGDs) can lead to the formation of orphan genes through mechanisms such as neofunctionalization and subfunctionalization, processes that allow duplicated genes to acquire novel functions or divide ancestral functions between them [93].

Studies have identified orphan genes in various plant species, including *Mikania micrantha*, which shows a significant number of unique gene families related to oxidative

phosphorylation, biosynthesis of unsaturated fatty acids, photosynthesis, and ether lipid metabolism. These pathways are crucial for growth, metabolism, and defense responses, highlighting the functional importance of orphan genes in adaptation to new environments [94]. The functional annotation of orphan genes poses challenges due to their lack of sequence similarity to known genes. Integrating transcriptomic data can provide insights into their roles as they can play a pivotal role in *Lgh*'s adaptive capacity for land habitats. Their unique functions, arising from genomic duplications, enable these plants to exploit new ecological niches and outcompete native species. Understanding the origin and function of orphan genes is crucial for unraveling the mechanisms underlying plant invasiveness, and for developing strategies to manage invasive species.

Among these orthologs, 40% (11,591) were annotated as hypothetical proteins. These genes could either be unique *Lgh* genes, or an 'overannotation' resulting from our annotation methods. If we remove them from the analysis, the proportion of *Lgh*-specific genes assigned to orthogroups drops to 13.7%, which is closer to values observed in other species. Very few genes in each genome belonged to family or order specific orthogroups.

## Synteny conservation in the Onagraceae family

The *Lgh* genome was aligned to the *E. ciliatum* genome with LASTZ to evaluate the synteny conservation (Figure 9). The synteny was often conserved between the two species. All chromosomes aligned almost entirely to *Lgh* scaffolds, showing an important sequence conservation between the two species. Twelve *E. ciliatum* chromosomes (out of 18) were aligned discontinuously with the *Lgh* genome with a big gap around the middle. These regions may correspond to centromeric regions which are known to contains a lot of repetitive elements, and which may be missing in the *Lgh* assembly [95]. These results are preliminary, however, as the analysis of synteny in incomplete plant genomes presents significant challenges that can compromise the accuracy and reliability of comparative genomic studies. In polyploid plants, where multiple subgenomes may exist, incomplete assemblies can lead to misinterpretation of syntenic regions, as gaps or misassembled contigs can disrupt the continuity of gene clusters. In addition, the presence of repetitive sequences and structural variations, common in plant genomes, further complicates the identification of syntenic blocks. These problems can lead to erroneous conclusions about chromosome evolution and gene function. To alleviate these difficulties, researchers are developing advanced computational tools and methodologies that integrate long-term sequencing data and use algorithms capable of handling incomplete assemblies. These approaches aim to improve the accuracy of synteny analysis and provide deeper insights into the genomic architecture of plants.

## CONCLUSION

Here, we report a draft genome assembly of the decaploid *Ludwigia grandiflora* subsp. *hexapetala* (*Lgh*), generated using both Illumina MiSeq and Oxford Nanopore GridION sequencing. Due to its high fragmentation and absence of chromosome-level scaffolds, structural analyses of this assembly should be approached with caution. Nevertheless, multiple lines of evidence, including k-mer analyses and RNA-based assessments, indicate that fragmentation does not compromise the completeness of non-repetitive, functionally informative regions, supporting the validity of this draft genome for gene annotation, comparative genomic, and evolutionary studies.



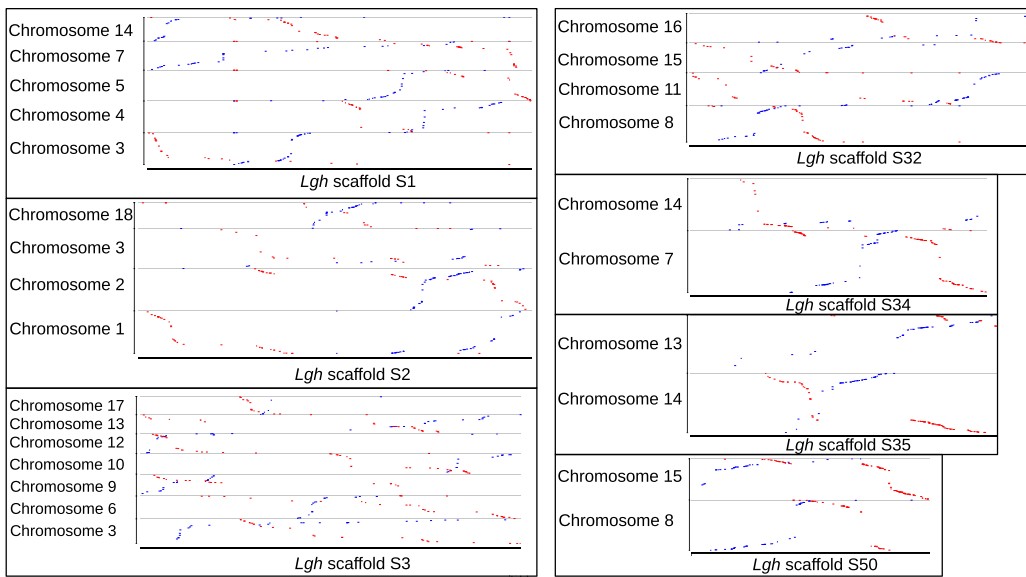

**Figure 9.** Dot plots of genome alignment between *Ludwigia grandiflora* subsp. *hexapetala* (*Lgh*) and *Epilobium ciliatum*. Only scaffolds > 1 Mb in *Lgh* were used.

This first draft genome represents a pivotal milestone as the inaugural reference for the Ludwigioideae subfamily. While completing sequencing to obtain chromosome-level assemblies would be valuable, this resource substantially fills a considerable gap in genomic data within the Onagraceae family, and more broadly within the Myrtales order, enhancing our ability to explore genetic diversity, infer phylogenetic relationships, and investigate the genetic basis of invasiveness. Our RNA-seq analysis further demonstrates the utility of this genome for functional genomics, providing a foundation for future studies of adaptation, evolution, and invasion biology.

## DATA AVAILABILITY

The genome assembly generated in this study and all sequencing reads can be accessed through EBI (BioProject number: PRJEB108588). The data sets supporting the results of this article are available in the GigaDB repository [96].

## LIST OF ABBREVIATIONS

BUSCO, Benchmarking Universal Single-Copy Orthologs; CDS, Coding-DNA sequence; GO, gene ontology; HMW, High-molecular-weight; *Lgh*, *Ludwigia grandiflora* subsp. *hexapetala*; LR, long reads; LTR, Long terminal repeat; NR, non-redundant; PCG, protein-coding gene; rRNA, ribosomal RNA; SMRT, single-molecule real-time; SR, short reads; tRNA, transfer RNA.

## DECLARATIONS

## Ethics approval and consent to participate

The authors declare that ethical approval was not required for this type of research.

## Consent for publication

Not applicable.

## Competing interests

The author(s) declare that they have no competing interests.

## Authors' contributions

DB and FB-H designed the project and defined the methodological approaches for which DB acquired funding. GD conducted the bioinformatics analyses, supervised by FB-H and DB. All authors wrote different parts of the manuscript and, together reviewed the manuscript.

## Funding

This research was funding by INRAE through the doctoral grant of Guillaume Doré.

## Acknowledgements

Not applicable.

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
