## [Reviewer Report]

Indicate in the comments box below whether you are happy with the changes made or if the manuscript is unacceptable.Comments on revised manuscriptI appreciate the authors’ careful and substantial revisions. The major concerns raised in the first round have been addressed in a constructive manner. The assembly remains highly fragmented due to low sequencing depth and the lack of high-molecular-weight DNA, and these intrinsic limitations cannot be fully resolved without additional data. However, the authors have now added k-mer–based completeness analyses, clarified the distinction between completeness and contiguity, acknowledged technical constraints explicitly, and provided independent synteny validation to reduce concerns about reference bias. Importantly, the gene annotation has been significantly improved through re-annotation with homology and RNA-seq support, leading to a more defensible gene set and reduced overprediction. Although the fundamental assembly limitations remain, the manuscript now transparently presents these constraints and positions the work appropriately as a draft genomic resource. In the context of GigaByte as a data-oriented journal, I consider the manuscript acceptable in its revised form.Indicate in the comments box below whether you are happy with the changes made or if the manuscript is unacceptable.Comments on revised manuscriptI appreciate the authors’ careful and substantial revisions. The major concerns raised in the first round have been addressed in a constructive manner. The assembly remains highly fragmented due to low sequencing depth and the lack of high-molecular-weight DNA, and these intrinsic limitations cannot be fully resolved without additional data. However, the authors have now added k-mer–based completeness analyses, clarified the distinction between completeness and contiguity, acknowledged technical constraints explicitly, and provided independent synteny validation to reduce concerns about reference bias. Importantly, the gene annotation has been significantly improved through re-annotation with homology and RNA-seq support, leading to a more defensible gene set and reduced overprediction. Although the fundamental assembly limitations remain, the manuscript now transparently presents these constraints and positions the work appropriately as a draft genomic resource. In the context of GigaByte as a data-oriented journal, I consider the manuscript acceptable in its revised form.

---

## [Editor Report]

Editor’s AssessmentThe manuscript is ready for formal acceptance.Editor’s AssessmentThe manuscript is ready for formal acceptance.

---

## [Reviewer Report]

Reviewer name and names of any other individual's who aided in reviewer Tao ShiDo you understand and agree to our policy of having open and named reviews, and having your review included with the published papers. (If no, please inform the editor that you cannot review this manuscript.)YesIs the language of sufficient quality?YesPlease add additional comments on language quality to clarify if needed
Are all data available and do they match the descriptions in the paper? NoAdditional CommentsThis assembly is not a functional genome but a fragmented gene catalog. The authors prioritized publishing a "first" genome for Ludwigioideae over technical rigor, ignoring their own admission that advanced methods (Hi-C/TELL-Seq) are essential for polyploids. Until HMW DNA is successfully extracted and sequenced with appropriate technologies, all downstream analyses remain fundamentally unreliable. The study's fundamental flaw lies in the failed extraction of high-molecular-weight (HMW) DNA, explicitly admitted by the authors: "our efforts to extract HMW DNA from Lgh have been unsuccessful" (Section 4.1). This resulted in severely compromised Nanopore long reads averaging only 4.9 kb (Fig. 2), far below the length required to span complex repetitive regions. For a decaploid genome (80 chromosomes) packed with rDNA clusters and Ty3-gypsy transposons (Fig. 6C), such short reads cannot resolve repeats, leading to massive assembly fragmentation (N50=13.5 kb, Table 2). Worse, the authors proceeded with sequencing despite knowing the DNA was "suboptimal," violating a core principle of long-read genomics. The annotation of 246,242 protein-coding genes (Fig. 3) is biologically implausible and reveals critical methodological errors: Compared to the decaploid Houttuynia cordata (139,087 genes in a larger 2.63 Gb genome), Lgh’s gene count is inflated by 77%. 116,869 genes (47%) are "hypothetical proteins" with no functional annotation, indicating either technical artifacts or contamination. Subcellular localization predictions (Fig. 5) show 34% of proteins targeting chloroplasts—double the typical plant rate —suggesting flawed algorithms? Low KEGG annotation (16% vs. 26% in Brassica) confirms poor functional reliability. There is intraspecific synteny or gene trees data to validate the ploid level of the assembed genome. Orthology analysis (Fig. 7) claims 42% species-specific "orphan genes"—an anomaly exceeding all 14 compared species (0.5-12.5%). However, 83% of these "orphans" are hypothetical proteins, suggesting artifactual gene calls. Worse, synteny maps (Fig. 8) are invalid due to reference-guided scaffolding bias: Scaffolds were forced onto Chamaenerion angustifolium chromosomes via RagTag, creating false synteny. Gaps in 11/16 chromosomes (Fig. 8) align with repetitive centromeres missing in the assembly. The authors acknowledge that synteny in incomplete genomes "can lead to erroneous conclusions", yet present these results as biological insights.Are the data and metadata consistent with relevant minimum information or reporting standards? See GigaDB checklists for examples <a href="http://gigadb.org/site/guide" target="_blank">http://gigadb.org/site/guide</a>YesAdditional CommentsIs the data acquisition clear, complete and methodologically sound?NoAdditional CommentsIs there sufficient detail in the methods and data-processing steps to allow reproduction?NoAdditional CommentsIs there sufficient data validation and statistical analyses of data quality? NoAdditional CommentsIs the validation suitable for this type of data?NoAdditional CommentsIs there sufficient information for others to reuse this dataset or integrate it with other data?NoAdditional CommentsAny Additional Overall Comments to the AuthorRecommendationReject (Unsound or Unusuable)

---

## [Reviewer Report]

Reviewer name and names of any other individual's who aided in reviewer Baocai HanDo you understand and agree to our policy of having open and named reviews, and having your review included with the published papers. (If no, please inform the editor that you cannot review this manuscript.)YesIs the language of sufficient quality?YesPlease add additional comments on language quality to clarify if needed
Are all data available and do they match the descriptions in the paper? YesAdditional CommentsAre the data and metadata consistent with relevant minimum information or reporting standards? See GigaDB checklists for examples <a href="http://gigadb.org/site/guide" target="_blank">http://gigadb.org/site/guide</a>YesAdditional CommentsIs the data acquisition clear, complete and methodologically sound?YesAdditional CommentsIs there sufficient detail in the methods and data-processing steps to allow reproduction?YesAdditional CommentsIs there sufficient data validation and statistical analyses of data quality? YesAdditional CommentsIs the validation suitable for this type of data?YesAdditional CommentsIs there sufficient information for others to reuse this dataset or integrate it with other data?YesAdditional CommentsAny Additional Overall Comments to the AuthorRecommendationAccept

---

## [Reviewer Report]

Reviewer name and names of any other individual's who aided in reviewer Hui ShangDo you understand and agree to our policy of having open and named reviews, and having your review included with the published papers. (If no, please inform the editor that you cannot review this manuscript.)YesIs the language of sufficient quality?YesPlease add additional comments on language quality to clarify if needed
Are all data available and do they match the descriptions in the paper? YesAdditional CommentsAre the data and metadata consistent with relevant minimum information or reporting standards? See GigaDB checklists for examples <a href="http://gigadb.org/site/guide" target="_blank">http://gigadb.org/site/guide</a>YesAdditional CommentsIs the data acquisition clear, complete and methodologically sound?YesAdditional CommentsIs there sufficient detail in the methods and data-processing steps to allow reproduction?YesAdditional CommentsThe methods and data processing steps are described in sufficient detail, allowing for reproducibility. However, the relatively low sequencing depth may affect the assembly quality, and it would be helpful to include specific tool versions to further enhance reproducibility.Is there sufficient data validation and statistical analyses of data quality? NoAdditional CommentsWhile the manuscript provides some data validation (e.g., BUSCO completeness, gene annotations, and repeat content), it lacks a more comprehensive statistical analysis of data quality. More detailed statistical methods for validating assembly quality and a more critical discussion of the validation metrics would improve the overall robustness of the study.Is the validation suitable for this type of data?YesAdditional CommentsThe validation methods used are generally suitable for the data type, but some improvements could be made, especially in terms of repeat content validation and gene annotation confirmation. Incorporating additional data types (such as RNA-seq) and using higher sequencing depth would provide more robust validation for the assembly and gene predictions.Is there sufficient information for others to reuse this dataset or integrate it with other data?NoAdditional CommentsThe work provides a valuable genomic resource for this underexplored group; however, the current assembly is highly fragmented, repeat content may be underestimated, and the number of predicted protein-coding genes is unusually high.These limitations make it challenging for others to use the dataset effectively.Any Additional Overall Comments to the AuthorThis manuscript reports the first draft nuclear genome assembly of Ludwigia grandiflora subsp. hexapetala, a decaploid invasive plant in the subfamily Ludwigioideae (Onagraceae). The authors used a combination of Illumina MiSeq short reads and Oxford Nanopore GridION long reads to produce a hybrid assembly, followed by gene prediction, functional annotation, and comparative genomic analyses within the Myrtales and Malvid clades. The work provides a valuable genomic resource for this poorly studied group, but the current assembly is highly fragmented, the repeat content is likely underestimated, and the number of predicted protein-coding genes is unusually high. These limitations should be more clearly stated in the abstract and discussion. There is some comments： 1. Title clarity The phrase “From Water to land” in the title could mislead readers into expecting mechanistic ecological or evolutionary studies on habitat transition, which is not the focus of this data release. Consider revising the title to reflect that this is a draft genome announcement rather than a functional adaptation study. 2. Assembly quality and limitations The genome is highly fragmented (111,219 contigs, N50 = 13.5 kb) with very low sequencing depth (6.5× Illumina + 1.6× Nanopore) for a complex polyploid plant. These limitations, along with the absence of chromosome-level scaffolds, should be emphasized in the abstract, results, and conclusion to manage reader expectations. The repeat content (8.5%) is much lower than expected for polyploid genomes. Additional evidence (e.g., read mapping to repeat databases) should be provided to support whether this is a biological feature or an artifact of incomplete assembly. Given the low sequencing depth and fragmented assembly, the reported BUSCO completeness score of 96.3% is unexpectedly high. This surprising result should be examined critically, as it may reflect overassembly of duplicated gene fragments or other annotation artifacts rather than true genome completeness. 3. Gene annotation concerns The predicted number of protein-coding genes (246,242) is exceptionally high compared to other polyploids, suggesting possible overannotation due to fragmented assembly and inclusion of spurious ORFs. Stronger filtering criteria (e.g., length thresholds, domain evidence) and annotation supported by RNA-seq would improve reliability. The high proportion of “orphan genes” in comparative analyses may largely result from annotation artifacts rather than genuine lineage-specific innovations. This possibility should be acknowledged explicitly. 4. Content relevance in Introduction The statement “further subdivided into six tribes: Epilobieae, Onagreae, Fuchsiaeeae, Clarkieae, Circaeeae, and Oenothereae” is tangential to the current study, as it does not directly relate to the focal taxon. Consider shortening or removing to improve focus. 5. Figures and data presentation In Figure 7, several genus abbreviations are ambiguous (multiple genera share the same two-letter code). Full Latin names are recommended, and “Lgh” should also be spelled out as Ludwigia grandiflora subsp. hexapetala to avoid confusion. In Figure 8, the meaning of the x- and y-axis labels is unclear. Please provide explicit axis descriptions (e.g., chromosome/scaffold identifiers, alignment positions) in both the figure and legend. 6. Comparative genomics and synteny analysis Since Chamaenerion angustifolium was used in scaffolding, the observed synteny with Lgh may be artificially inflated. This limitation should be discussed in the synteny section. 7. Some Minor Comments Method descriptions, particularly in the assembly and decontamination sections, are overly detailed. A summarized pipeline diagram would make the workflow easier to follow. Citation styles are inconsistent (mix of DOI links, URLs, and bare accession numbers). Please unify according to journal guidelines. Minor typographical issues: e.g., “ChlorophytaPhylums” should be “Chlorophyta phyla”.RecommendationMajor Revision